# Structure Size as Confounder in Uncertainty Based Segmentation Quality Prediction

**Kai Geißler**[1]                                   KAI.GEISSLER@MEVIS.FRAUNHOFER.DE
[1] *Fraunhofer Institute for Digital Medicine MEVIS, Bremen, Germany*
**Jochen G. Hirsch**[1]
**Stefan Heldmann**[1]
**Hans Meine**[1]

**Editors:** Accepted for publication at MIDL 2024

## Abstract

Various uncertainty estimation methods have been proposed for deep learning-based image segmentation models. An uncertainty measure is treated useful if it can be used to accurately predict segmentation quality. Therefore, structure-wise uncertainty measures are frequently correlated with measures like the Dice score. However, it is known that the Dice score highly depends on the size of the structure of interest. It is less well-known that popular structure-wise uncertainty measures also correlate with structure size. Therefore, the structure size acts as confounding variable when trying to quantify the performance of such uncertainty measures via correlation. We investigate this for the popular uncertainty measures structure-wise epistemic uncertainty, mean pairwise Dice and volume variation coefficient based on test-time-augmentation, Monte Carlo Dropout and model ensembles. We propose to use a partial correlation coefficient to address structure size as confounding variable and arrive at lower correlation estimates which better reflect the true relationship between segmentation quality and structure-wise uncertainty.

**Keywords:** Uncertainty Quantification, Medical Image Segmentation

## 1. Introduction

Estimation of model uncertainty in deep learning based image segmentation can be done with various proposed measures such as mean predictive entropy (Gal et al., 2017), mutual information (Kendall and Gal, 2017), volume variation coefficient (VVC) (Roy et al., 2018), or mean pairwise Dice (MPD) (Roy et al., 2018). Given a specific model input, there is also a choice of methods to produce a set of outputs, like test time augmentation (TTA) (Wang et al., 2019), Monte Carlo dropout (MCD) (Gal and Ghahramani, 2015; Gal et al., 2017) or model ensembles (Lakshminarayanan et al., 2017), which we will call uncertainty sources. Uncertainty measures then translate such sets of outputs into an uncertainty prediction, i.e. an estimate of the likelihood of making a wrong prediction. In order to assess an uncertainty measure, it is commonly correlated with a measure for the prediction quality. In the case of image segmentation, this is usually the Dice score, which is known to be higher in general for larger structures (Reinke et al., 2021). If model uncertainty was also influenced by structure size, then the structure size would be a confounding variable, potentially tainting correlation estimates.

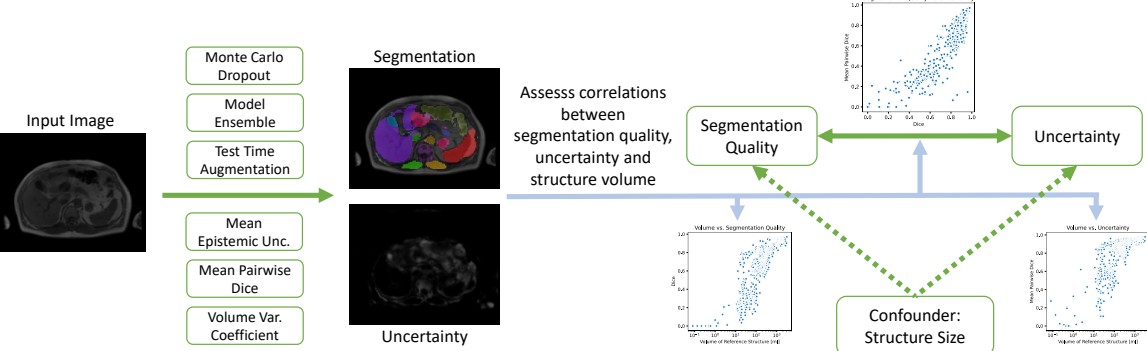

Figure 1: A proper uncertainty measure for structure segmentation quality should be well correlated with the segmentation quality. We assess the confounding effect of the structure volume on this relationship.

In this work, we analyse this relationship for the task of segmenting diverse anatomical structures in whole body MRI. Our contributions are threefold:

- We show empirically that the popular uncertainty measures structure-wise epistemic uncertainty, MPD and VVC all correlate notably with structure size.

- To compute correlation estimates that correct for this confounding variable, we propose to use partial Spearman's rank correlation (Liu et al., 2018) and assess its changes to the correlation estimates and to the resulting ranking of uncertainty measures.

- We also confirm that the average symmetric surface distance has a lower correlation with structure size and is therefore less severely affected its confounding nature.

Our results are achieved in the context of domain transfer between computed tomography (CT) which is our source domain and magnetic resonance imaging (MRI) which is our target domain. A segmentation model is trained on the TotalSegmentator dataset (Wasserthal et al., 2023), a large collection of CT images with masks for anatomical structures. During training it utilizes GIN augmentation (Ouyang et al., 2023) to allow domain transfer. It is evaluated on a subset of the MR Imaging Study within the German National Cohort Study (Bamberg et al., 2015) for which segmentation masks are manually created by radiological technologists.

## 2. Related Work

Different ways have been proposed to estimate the uncertainty of deep neural networks, like model ensembles (Lakshminarayanan et al., 2017), test time augmentation (Wang et al., 2019), using the raw model outputs (Smailagic et al., 2018), Monte Carlo dropout (Gal and Ghahramani, 2015, 2016) or deterministic methods (Liu et al., 2020a). In addition there are uncertainty aware models using variational autoencoders to learn the model uncertainty (Kohl et al., 2018) and models based on Dempster-Shafer theory (Huang et al., 2022) or

Table 1: Description of study population using mean $\pm$ SD for continuous variables

| Cases | Study Centers | Sex | Age [Years] | Weight [kg] | Height [cm] |
|---|---|---|---|---|---|
| 30 | 5 | 15 male, 15 female | $45.7 \pm 13.6$ | $86.9 \pm 21.4$ | $173.9 \pm 13.2$ |

subjective logic (Zou et al., 2022) using evidential deep learning to incorporate the model uncertainty. There are also models which try to disentangle the uncertainty coming from different possible uncertainty sources (Shaw et al., 2021).

Mehta et al. (2022) conducted the QU-BraTS challenge assessing the uncertainty prediction performance of 14 participating teams and proposing a novel metric to evaluate uncertainty maps. Camarasa et al. (2021) evaluated how to best calculate voxel-wise uncertainty maps in multi-class segmentation settings.

Based on sampled model outputs, there are different ways to compute uncertainties for each structure. One way is to compute the voxel-wise uncertainty of the image, such as entropy, variance or variation ratio over the samples (Gal et al., 2017) and average it over the predicted structure mask. For segmentation tasks, there are also structure level measures which can be computed directly from a set of masks, like the volume variation coefficient and the mean pairwise Dice score between the predictions (Roy et al., 2018).

Kendall and Gal (2017) introduced an uncertainty decomposition into two components: Aleatoric uncertainty captures uncertainty inherent to the data, like measurement or labeling errors, while epistemic uncertainty captures the uncertainty in the model parameters by computing the mutual information (Mukhoti and Gal, 2018).

Many publications use correlations between segmentation quality and uncertainty measures to compare different methods (Roy et al., 2018; Wang et al., 2019; Hoebel et al., 2020; Lin et al., 2022a,b; Sahlsten et al., 2023) while others report correlations on their own to assess the quality of the uncertainty measure (Hiasa et al., 2019). Hoebel et al. (2020) found no significant correlation between uncertainty measures and structure size in their experiments on lung nodule segmentation in CT scans, computing uncertainties based on MCD and model ensembles using structure-wise mean entropy over the predicted samples, VVC and MPD. Furthermore, Wang et al. (2019) claim that the VVC is agnostic to structure size.

In contrast to this, we find non-negligible correlations between the structure size and the uncertainty measures structure-wise epistemic uncertainty, VVC and MPD based on model ensembles, MCD and TTA in our experiments.

## 3. Experiments

### 3.1. Data

For model training the publicly available TotalSegmentator dataset is used. It contains 1204 CT images with 104 annotated structures (Wasserthal et al., 2023), accounting for 27 organs, 59 bones, 10 muscles, and eight vessels.

Our evaluation dataset consists of 30 selected cases out of 10828 whole body MR volumes obtained as part of the MR Imaging Study within the German National Cohort Study (GNC, 2014-2019) (Bamberg et al., 2015) from volunteers. The data were acquired on MAGNETOM Skyra 3 T (Siemens Healthineers, syngo VD13C) systems. The MRI sequence is a two-point Dixon volumetric interpolated breath-hold examination (VIBE) with T1 weighting. Axial slices were acquired with a $320 \times 260$ in-plane matrix (resolution $1.4 \times 1.4\,mm^2$) and a slice thickness of $3\,mm$. The volume consists of four acquired table positions with a total of 316 slices. The test cases are selected to span a diverse sample of the subject distribution with regard to study center, sex, age, weight and height. Details of the study population are depicted in Table 1.

Three radiological technologists annotate segmentation masks for 28 anatomical structures, comprising 8 abdominal organs (liver, spleen, kidneys, stomach, pancreas, adrenal glands), 5 thoracic organs (heart, lungs, esophagus, trachea), 12 bones (scapulas, claviculas, hips, sacrum, femurs, vertibrae L1-5, T1-12 and C3-7), 2 muscles (autochthon) and one vessel (aorta) on these cases. The number of annotated structures per structure type is shown in Table 6 in the appendix. The structure types are chosen as a subset of the TotalSegmentator structures. This variety was selected to be able to draw conclusions about anatomical structures of various shapes and sizes.

## 3.2. Segmentation Model

A segmentation model is trained from scratch on the TotalSegmentator data using our re-implementation of the nnU-Net framework (Isensee et al., 2021). As training configuration, we approximate the TotalSegmentator low resolution ($3\,mm$) model. The images are resampled to a voxel size of $3 \times 3 \times 3\,mm^3$, a patch size of $80 \times 80 \times 80$ is used and the nnU-Net non-CT normalization is performed. The models are trained for 250,000 iterations with a batch size of 2. The model used for MCD has a dropout layer with dropout rate 0.1 after each block of convolution, normalization and non-linearity in the up- and downpath. For basic data augmentation we utilize the batch generators library (Isensee et al., 2020).

In addition all models are trained using GIN data augmentation (Ouyang et al., 2023) to allow for domain transfer between CT and MR images. In GIN, a convolutional neural network (CNN) applies a random non-linear intensity value transformation on the training patches. Therefore, the model needs to focus more on the shape and less on the intensity values of structures and generalizes better to different imaging modalities and contrasts. We re-implement GIN augmentation. In our experiments, GIN augmentation is applied with a probability of 0.9 to each patch. The CNN for the augmentation has 4 layers with 2 channels each and uses ReLU activations. In each layer of the GIN augmentation network a random $1 \times 1 \times 1$ convolution is sampled. As we train our models on a low resolution we decided to use only $1 \times 1 \times 1$ convolutions to avoid too much smoothing that could arise from larger filters.

## 3.3. Uncertainty Quantification

We asses three alternative uncertainty sources to predict a set of different predictions for the same input image: model ensembles, TTA and MCD. For model ensembles one trains a set of deep learning models to obtain models with different learned weights for the prediction

task. In MCD this process is approximated by training only one model with dropout layers (Srivastava et al., 2014); in our experiments, we use spatial dropout (Tompson et al., 2015) which is more suited for convolutional neural networks. During inference, the dropout layers are then activated to use the model as Bayesian neural network that approximates the model posterior with different predictions for each dropout layer configuration. In TTA, there is only one fixed model, but for each inference pass the input image is augmented with either reversible geometric augmentations or intensity transformations to obtain a set of different predictions for the same input. In our experiments 3D rotations, scaling, additive and multiplicative brightness, contrast transformation and gamma transformation are used. These are inspired by the data augmentations used in the batch generators library.

The epistemic uncertainty (Kendall and Gal, 2017; Mukhoti and Gal, 2018) is first computed for each voxel by Equation 1, where $p_{ct}(x)$ is the predicted probability for class $c$, voxel $x$ and sample $t$. $T$ is the number of samples used for TTA, MCD and model ensembles, which is set to 10 in our experiments.

$$\mathbb{I}(x) = -\sum_c \left( \frac{1}{T} \sum_t p_{ct}(x) \right) \log \left( \frac{1}{T} \sum_t p_{ct}(x) \right) + \frac{1}{T} \sum_t \sum_c p_{ct}(x) \log p_{ct}(x) \qquad (1)$$

In order to then get to a structure-wise epistemic uncertainty, we average the epistemic uncertainty over all voxels which are predicted as this structure.

The MPD (Roy et al., 2018) for class $c$ is computed as the mean of the pairwise Dice scores of all predicted segmentation masks (Eq. 2). $m_{ct}$ is the binary mask for class $c$ and sample $t$.

$$\mathrm{MPD}_c = \frac{2}{T(T-1)} \sum_{i>j} \mathrm{Dice}(m_{ci}, m_{cj}) \qquad (2)$$

Finally, the VVC (Roy et al., 2018) for class $c$ is computed as the variance of the predicted structure volumes divided by their mean (Eq. 3).

$$\mathrm{VVC}_c = \mathrm{Var}_t(\mathrm{vol}(m_{ct})) \,/\, \mathrm{E}_t(\mathrm{vol}(m_{ct})) \qquad (3)$$

### 3.4. Partial Spearman's Rank Correlation

When computing the correlation between two variables X and Y, which are both highly correlated with a third variable Z, the correlation between X and Y gets tainted by the confounding nature of Z and potentially overestimated. In our case X and Y are the segmentation quality and structure-wise uncertainty while the potential confounder Z is the structure size. Partial correlation coefficients allow to remove the confounding effect of Z when assessing the correlation between X and Y, providing a better estimate of their actual relationship.

As we do not observe a linear relationship between our segmentation quality metrics and uncertainty measures, we cannot use partial Pearson's correlation for our purpose, but have to rely on rank based correlations. Kendall (1942) defined a partial correlation based on Spearman's rank correlation, but stated that it is hard to theoretically justify. It was also criticized because it is nonzero even under conditional independence of the correlated variables given the controlling variable (Korn, 1984). Therefore we propose to use the partial Spearman's rank correlation defined by Liu et al. (2018) which they base on probability scale residuals. To compute it we use the R package PResiduals (Liu et al., 2020b).

Table 2: Correlation between uncertainty measures (left) or segmentation quality (right) and structure volume computed over all structures. Signs of some quantities are switched to allow for easier comparison of correlations.

|  | -Ep. Unc. | MPD | -VVC | Dice | -ASSD |
|---|---|---|---|---|---|
| TTA | 0.627 | 0.776 | 0.782 | 0.804 | 0.246 |
| Ensemble | 0.807 | 0.757 | 0.735 | 0.776 | 0.193 |
| MCD | 0.641 | 0.769 | 0.850 | 0.814 | 0.134 |

Table 3: Correlation between uncertainty measures and segmentation quality. Bold values mark the best method/measure combination for each segmentation quality metric.

|  | 1-Dice vs. Ep. Unc. | 1-Dice vs. 1-MPD | 1-Dice vs. VVC | ASSD vs. Ep. Unc. | ASSD vs. 1-MPD | ASSD vs. VVC |
|---|---|---|---|---|---|---|
| TTA | 0.845 | 0.936 | 0.850 | 0.635 | 0.586 | 0.562 |
| Ensemble | 0.908 | 0.941 | 0.801 | 0.746 | **0.788** | 0.694 |
| MCD | 0.837 | **0.958** | 0.907 | 0.660 | 0.577 | 0.520 |

## 4. Results

### 4.1. Evaluation Across All Structures

At first, we consider Spearman's correlation between structure volume and uncertainty, as well as between structure volume and segmentation quality. The results across all structures and patients are presented in Table 2. We observe strong correlation for all tested uncertainty measures and for the Dice score, which is contrary to the observations of Hoebel et al. (2020) who observed no significant correlations between structure volume and uncertainty. For the average symmetric surface distance (ASSD), we observe only weaker correlations.

The correlation between uncertainty measures and segmentation quality is shown in Table 3. We observe moderate to high correlations in all cases, ranging from 0.520 to 0.958, again higher when using the Dice score as segmentation quality measure.

As we observed before, there is a high correlation between both the segmentation quality and structure volume as well as uncertainty and structure volume. Therefore, one can see the structure volume acting as confounding variable between segmentation quality and uncertainty. To correct for this, we use the partial Spearman's rank correlation between segmentation quality and uncertainty, accounting for the structure volume. It removes the influence of structure volume from the correlation between segmentation quality and uncertainty. We show the results in Table 4. One can observe that the correlations drop in all cases, sometimes considerably by up to 0.260. We can also compare Table 3 and Table 4 to check if the ranking of different uncertainty methods (column wise) or uncertainty measures (row wise) changes when switching from correlation to partial correlation. This

Table 4: Partial correlation between uncertainty measures and segmentation quality controlling for structure volume. Differences to regular correlation (Table 3) are displayed in gray, measures that rank better (worse) within their column are marked with △ (▽) and uncertainty sources that changed rank within their row with ▲(▼). Bold values mark the best method/measure combination for each segmentation quality metric.

| | 1-Dice vs. Ep. Unc. | 1-Dice vs. 1-MPD | 1-Dice vs. VVC | ASSD vs. Ep. Unc. | ASSD vs. 1-MPD | ASSD vs. VVC |
|---|---|---|---|---|---|---|
| TTA | ▲0.736 | **0.820**△ | ▼0.665 | 0.607 | 0.553 | 0.508 |
| △ | -0.109 | -0.116 | -0.185 | -0.028 | -0.033 | -0.054 |
| Ensemble | ▲0.792 | ▼0.726▽ | 0.541 | ▲**0.678** | ▼0.629 | 0.534 |
| △ | -0.116 | -0.215 | -0.260 | -0.068 | -0.159 | -0.160 |
| MCD | ▲0.698 | 0.799▽ | ▼0.675 | 0.638 | 0.513 | 0.443 |
| △ | -0.139 | -0.159 | -0.232 | -0.022 | -0.064 | -0.077 |

is important, because it means that we potentially arrive at different conclusions about the utility of uncertainty methods/measures when we assess them either with the correlation or partial correlation. We observe that the ranking of different methods is relatively stable (only one column changes rankings) while for the different measures the rankings differ more (all three rows changing for Dice and one row changing for ASSD).

From these observations we can conclude that the structure volume has a strong confounding effect on the correlation between uncertainty measures and the Dice score and a weak confounding effect on the correlation between uncertainty measures and ASSD. The partial correlation allows to address this issue and arrive at correlation estimates less tainted by this confounding variable.

### 4.2. Evaluation per Structure Type

In the previous section we observed a strong correlation between structure-wise uncertainty measures and structure volume when computing it across a wide variety of structure types. All these structure types differ in their typical volume (cf. Table 6 in the appendix), which is why the structure type itself could be the confounding variable, with the structure volume being only an intermediate dependent variable on that. So the remaining question is if the structure volume is the true confounding variable or if it is actually the structure type.

Considering correlation between structure size and uncertainty or segmentation quality, the median, $1^{st}$ and $3^{rd}$ quartiles over all 28 structure types are shown in Table 5. The correlations between structure volume and uncertainty are weaker when evaluated per structure type, but still many structure types show correlations that deviate considerably from 0.

When correlating the uncertainty measures with the Dice score per structure type, the medians of the distribution of correlations move towards 0 in all nine cases we are assessing when switching from correlation to partial correlation. The lower and upper quartiles also move towards 0 in eight of nine cases each. For the ASSD the medians

Table 5: Distribution over correlation between uncertainty measures or segmentation quality and structure volume over structure types (median [$1^{st}$ quartile, $3^{rd}$ quartile])

|  | -Ep. Unc. | MPD | -VVC | Dice | -ASSD |
|---|---|---|---|---|---|
| TTA | 0.23 [-0.02,0.37] | 0.29 [0.17,0.42] | 0.17 [0.08,0.36] | 0.51 [0.27,0.68] | 0.25 [0.13,0.35] |
| Ensem. | 0.23 [0.13,0.51] | 0.30 [0.07,0.53] | 0.29 [0.06,0.44] | 0.51 [0.37,0.69] | 0.25 [0.18,0.37] |
| MCD | 0.16 [0.06,0.30] | 0.19 [0.04,0.38] | 0.20 [0.11,0.35] | 0.58 [0.29,0.73] | 0.15 [0.00,0.40] |

and quartiles sometimes rise and sometimes fall when switching from correlation to partial correlation. Figure 2 in the appendix summarizes the distribution over correlations and partial correlations per structure type and Figure 3 and Figure 4 in the appendix provide all the individual correlations. This shows that the confounding nature of structure volume is still present when assessing individual structure types and the partial correlation should be used. When correlating with the ASSD this effect is much less severe. This highlights its usefulness to assess the quality of uncertainty measures, as it is less effected by the confounding nature of structure volume.

### 4.3. Statistical Testing for Structure-wise Evaluation

To test if partial correlation also leads to consistent reductions when considering structure types individually, we perform a one-sided Wilcoxon signed-rank test over all types. The threshold for significance is set to $p < 0.0028$ based on Bonferroni correction to achieve a family wise error of 0.05. We find significance for TTA using either MPD or VVC as uncertainty measure and Dice score as segmentation quality metric.

## 5. Conclusion

We evaluated the influence of structure volume as confounding variable on the correlation between model uncertainty and segmentation quality in medical image segmentation. It was confirmed that the Dice score has a strong correlation with structure volume, and in contrast to prior work, we also found model uncertainty to be strongly correlated with structure volume. This effect was evaluated both across various structure types as well individually per structure type. It appears stronger in the first case, but is still visible per structure type.

To counteract this issue, we propose to use the partial correlation coefficient when correlating segmentation quality and model uncertainty, which removes the confounding effect of the structure volume from the correlation estimate. In addition, we observe that the average symmetric surface distance suffers less from this issue, as it has only a very weak correlation with structure volume. This makes it a suitable segmentation quality measure that can be used in addition to the Dice score to assess the performance of uncertainty measures.

## Acknowledgments

This work was supported within the Fraunhofer and DFG transfer programme. We thank Sophia Winkler, Christiane Engel and Andrea Koller for their help with creating manual annotations. We thank Ole Schwen and Max Westphal for helpful discussions. We also acknowledge the German National Cohort study for collecting the data on which we performed our evaluation.

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

## Appendix A. Number of Masks and Volumes per Structure Type

Table 6: Volume of individual structure types

| Structure Type | Number of masks | Volume [ml] (mean ± std. dev.) |
|---|---|---|
| Adrenal Gland Left | 8 | 1.13 ± 0.93 |
| Adrenal Gland Right | 6 | 0.46 ± 0.37 |
| Aorta | 17 | 159.86 ± 54.09 |
| Autochthon Left | 18 | 513.04 ± 193.42 |
| Autochthon Right | 16 | 487.95 ± 187.71 |
| Clavicula Left | 24 | 24.49 ± 9.67 |
| Clavicula Right | 24 | 23.89 ± 8.22 |
| Esophagus | 17 | 23.37 ± 6.34 |
| Femur Left | 18 | 341.03 ± 52.93 |
| Femur Right | 17 | 337.94 ± 48.76 |
| Heart | 24 | 495.06 ± 173.17 |
| Hip Left | 25 | 299.66 ± 98.55 |
| Hip Right | 25 | 306.24 ± 78.32 |
| Kidney Left | 18 | 145.69 ± 36.08 |
| Kidney Right | 18 | 143.43 ± 34.84 |
| Liver | 26 | 1660.21 ± 410.92 |
| Lung Left | 25 | 1715.26 ± 437.92 |
| Lung Right | 25 | 2046.60 ± 485.89 |
| Pancreas | 17 | 73.56 ± 19.35 |
| Sacrum | 24 | 158.62 ± 42.40 |
| Scapula Left | 24 | 71.61 ± 17.90 |
| Scapula Right | 24 | 78.51 ± 27.39 |
| Spleen | 18 | 211.46 ± 71.36 |
| Stomach | 18 | 222.13 ± 140.41 |
| Trachea | 18 | 27.16 ± 9.63 |
| Urinary Bladder | 19 | 154.18 ± 121.17 |
| Vertebrae C | 14 | 18.66 ± 7.53 |
| Vertebrae L | 17 | 229.63 ± 70.20 |
| Vertebrae T | 17 | 242.51 ± 94.14 |

# Appendix B. Distribution of Correlation between Uncertainty and Segmentation Quality per Structure Type

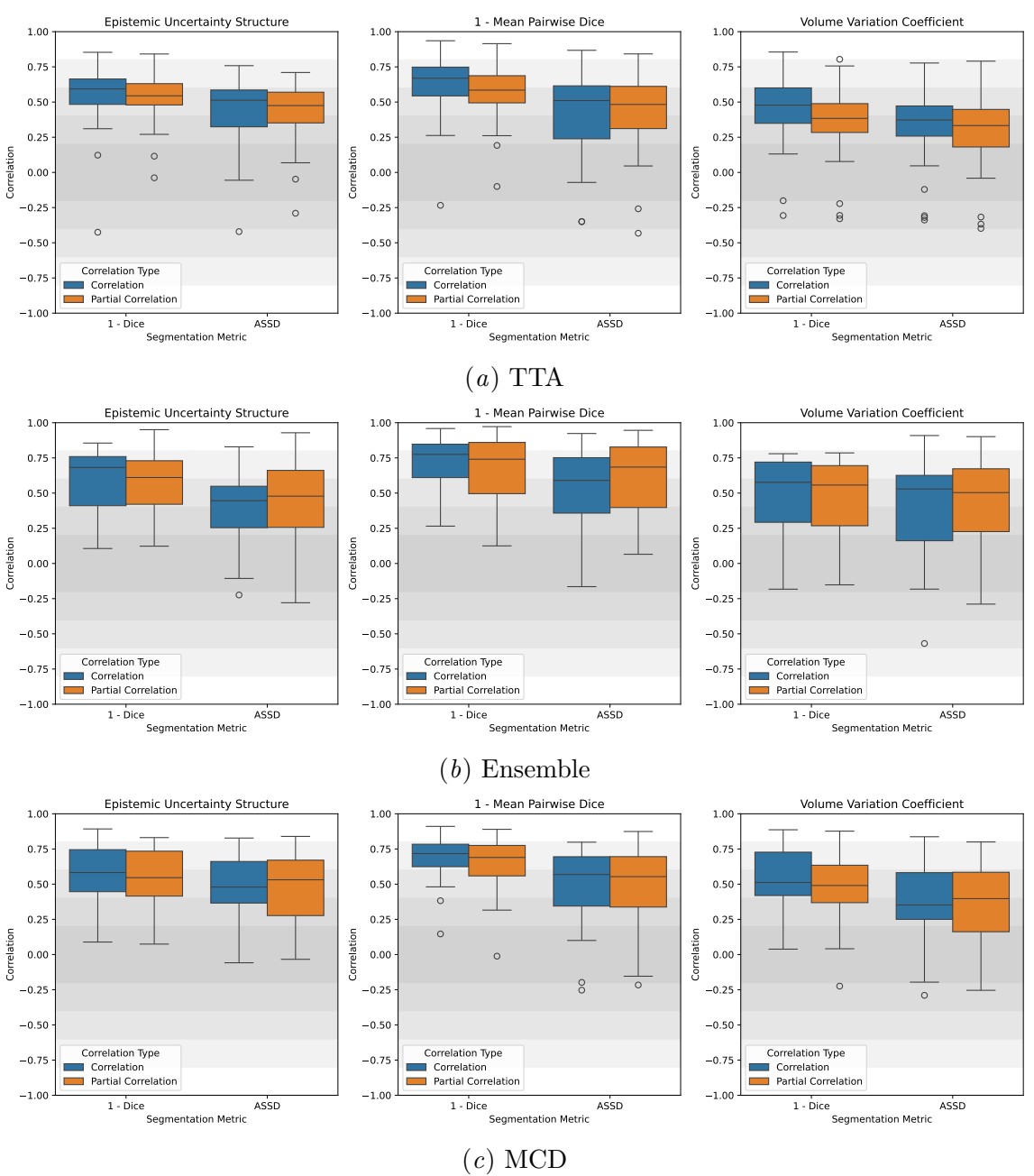

Figure 2: Distribution over (partial) correlation between segmentation quality and uncertainty for individual structure types.

## Appendix C. Correlation of Uncertainty and Segmentation Quality per Structure Type

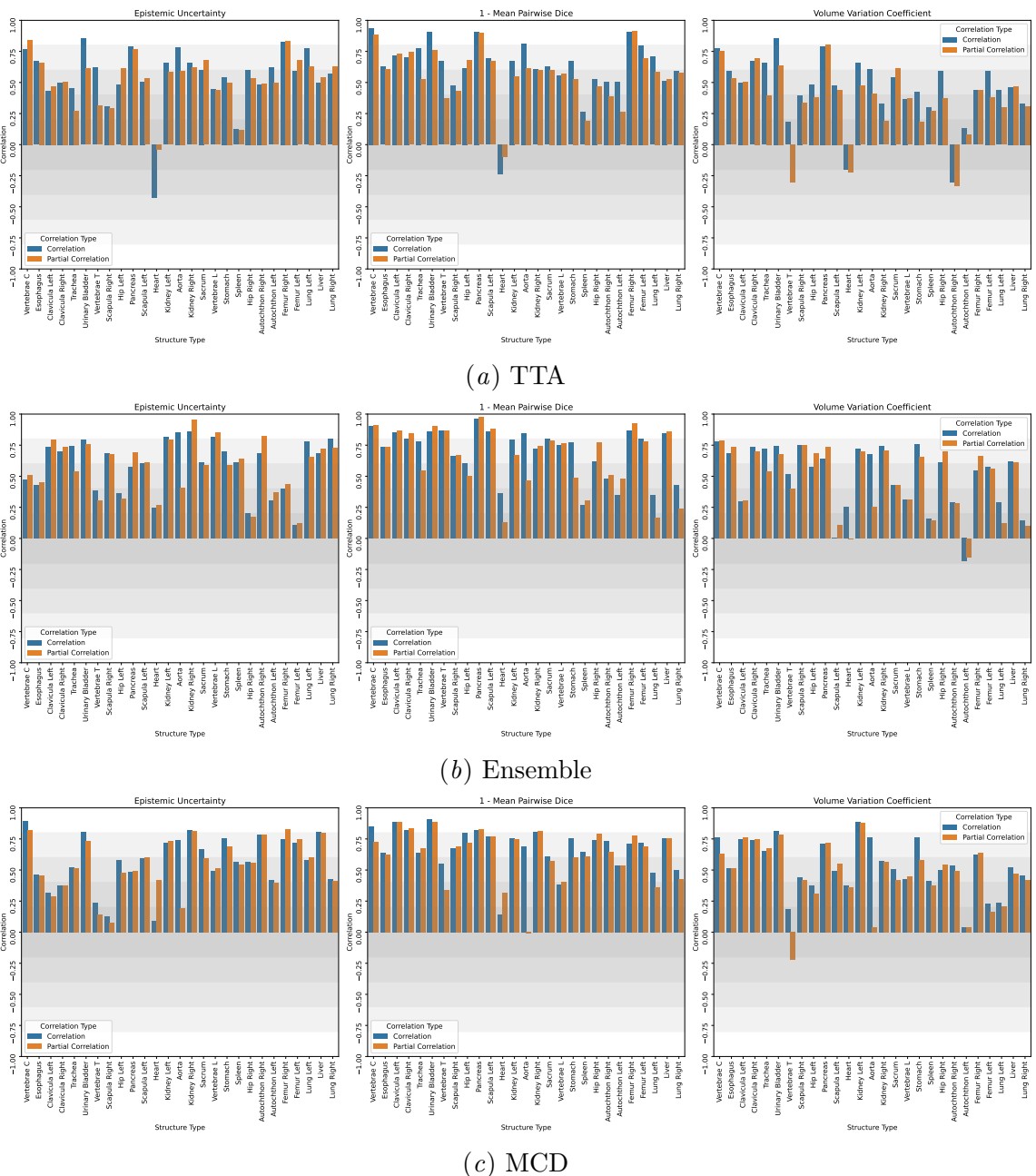

(*a*) TTA

(*b*) Ensemble

(*c*) MCD

Figure 3: (Partial) correlation between 1 - Dice and uncertainty for individual structure types.

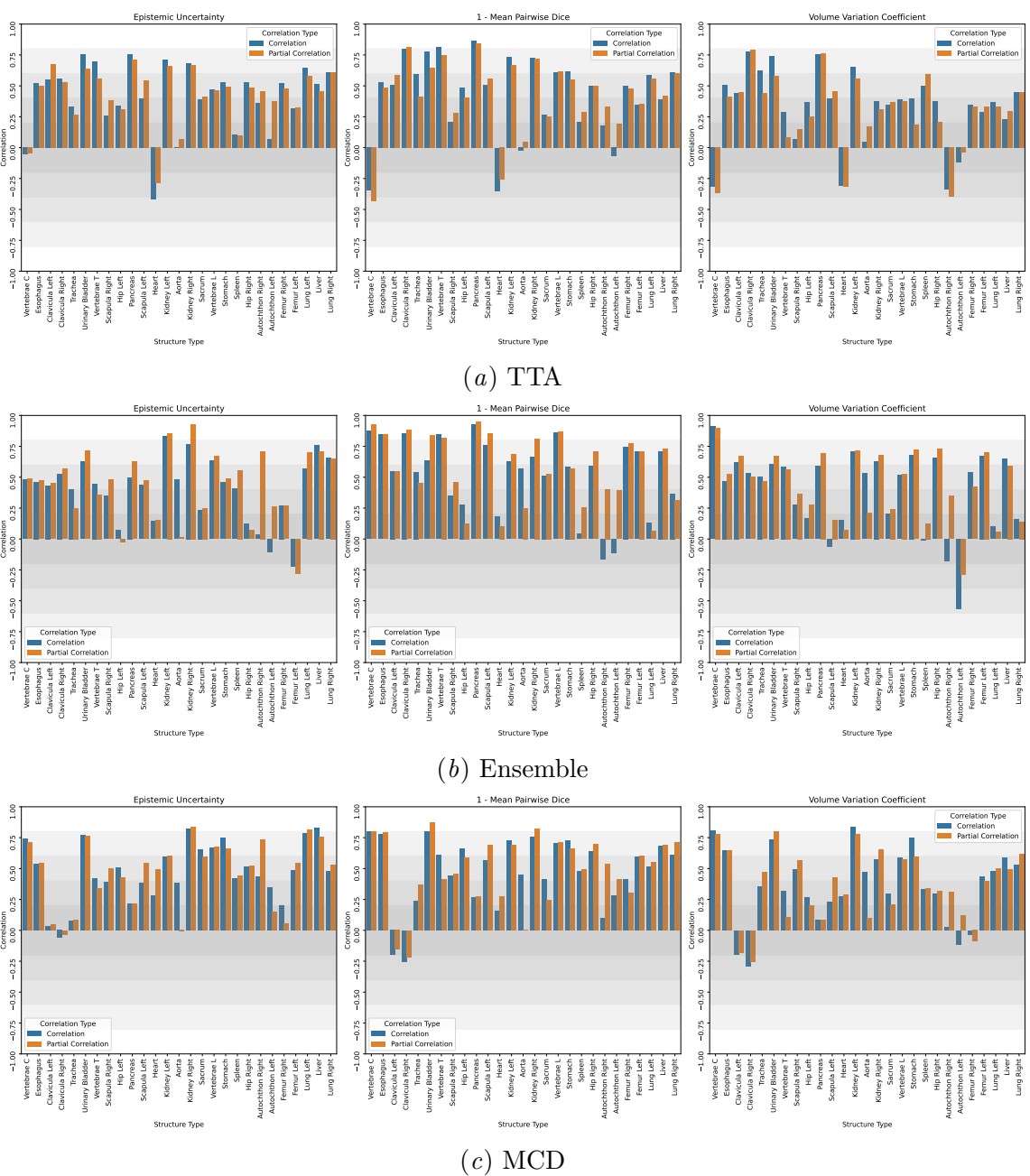

Figure 4: (Partial) correlation between average symmetric surface distance and uncertainty for individual structure types.

