# OpenReview forum: "Structure Size as Confounder in Uncertainty Based Segmentation Quality Prediction"
_MIDL.io/2024/Conference — MIDL 2024 Poster_

### Official Review · Reviewer_uNpG · 2024-02-21

**Confidence:** 3
**Preliminary Rating:** 2
**Recommendation:** Poster
**Final Rating:** 3.5

**Summary:**

This paper investigates the interesting point for the popular uncertainty measures structure-wise epistemic uncertainty, mean pairwise Dice and volume variation coefficient based on test-time-augmentation, Monte Carlo Dropout and model ensembles. They propose to use a partial correlation coefficient to address structure size as confounding variable and arrive at lower correlation estimates. The research in this paper has really aroused my interest. However, the uncertainty estimation methods in this paper are not sufficiently investigated and the experiments are insufficient. If the author can better respond to the shortcomings, I may change the grade.

**Strengths:**

This paper investigates the interesting point for the popular uncertainty measures structure-wise epistemic uncertainty, mean pairwise Dice and volume variation coefficient based on test-time-augmentation, Monte Carlo Dropout and model ensembles. They propose to use a partial correlation coefficient to address structure size as confounding variable and arrive at lower correlation estimates.

**Weaknesses:**

1. Contributions to this article are not listed in the intro.
2. The uncertainty estimation methods compared in this paper are insufficient.
3. The output in Figure 1 requires a clear finding and conclusion.
4. Insufficient research on uncertainty estimation.
5. How about transformer and SAM?
6. More indicators on uncertainty estimation, such as ECE and UEO[1], are not introduced.

[1] Assessing reliability and challenges of uncertainty estimations for medical image segmentation

**Detailed Comments:**

1. List your contributions in the intro.
2. Compare more uncertainty estimation methods [1-3].
3. Specify a finding and conclusion in the output in Figure 1.
4. Add more work related to uncertainty estimation [1-3].
5. How is the performance of this study when transformer and SAM based methods are introduced as the basic network?
6. Introduce more indicators on uncertainty estimation, such as ECE and UEO[4].
[1] A Probabilistic U-Net for Segmentation of Ambiguous Images
[2] TbraTS: trusted brain tumor segmentation
[3] Lymphoma segmentation from 3D PET-CT images using a deep evidential network
[4] Assessing reliability and challenges of uncertainty estimations for medical image segmentation

**Justification Of Final Rating:**

Thanks to the author's positive improvements, I believe several issues that were concerning to me have been addressed.
They extended the related works.
They adapted Figure 1.
But I still reserve the opinion that some methods should be compared to show that the authors' findings apply to a variety of models and measures.
As a result, I've decided to adjust my rating accordingly.

**Justification Of The Preliminary Rating:**

The basis of the preliminary rating mainly comes from the following aspects: the review of uncertainty estimation methods, uncertainty estimation indicators, the application and experiment of uncertainty estimation methods.

**Questions To Address In The Rebuttal:**

See Weaknesses and detailed Comments.

**Special Issue:**

No

---

> ### Author Response · Authors · 2024-03-17
> **We thank you for your valuable feedback and adressed it in our draft as follows:**
>
> 1. We listed our contributions with bullet points for clarity.
> 2. We adapted Figure 1 to better represent our main message, as you were not the only reviewer pointing out that it needed to be improved.
> 3. Our paper is not about finding the best uncertainty method, but we want to point out the methodological pitfall of structure size being a confounding variable in this setting (which is hopefully also more clear from the better Figure 1 and its new caption).
> 4. We extended the related work section with more uncertainty estimation methods.
> 5. As already written above, our contribution is not to assess and compare as many models or uncertainty measures as possible, and we hope this becomes much more clear now in our revised paper. We have merely selected a number of uncertainty estimation methods to show that the problem we highlight and address does not only affect a single one, but that our findings hold across a variety of models and measures. We expect other base networks to behave similarly as the methods we looked at, as they are based on the creation of a set of predictions from model ensembles, TTA or MCD and deriving uncertainties from these predictions, but we did not test all base networks from the literature yet.
> 6. While the indicators you proposed (expected calibration error (ECE) and uncertainty-error overlap (UEO)) are valuable measures for assessing uncertainty estimation methods, they are not directly suited to evaluate structure-wise uncertainty predictions, as they act on the uncertainty maps and not on the structure-wise uncertainties. As we are looking at the latter in this work, we did not include an evaluation of these metrics.

---

### Official Review · Reviewer_hMjP · 2024-02-28

**Confidence:** 4
**Preliminary Rating:** 3
**Recommendation:** Poster
**Final Rating:** 3.5

**Summary:**

The paper proposes to investigate structure size as a confounder variable in utilizing uncertainty estimates for segmentation quality prediction. To correct for this, they propose to use partial Spearman’s rank correlation. The experimental results indicate that there is a strong correlation between uncertainty, dice, and structure size, while that is not the case for distance-based segmentation quality measure.

**Strengths:**

* The paper proposes to evaluate the correlation between structure size, dice score, ASSD, and different uncertainty measures. The experiments are thorough.
* This is a motivating and timely evaluation of the correlation between structure size and uncertainty measures.
* The majority of the paper is well written.

**Weaknesses:**

* Missing citations in the relevant work section [1] [2] [3].
* It took a while to understand that the MR data was only utilised for testing purposes. Maybe the authors want to clarify this further.
* It is not clear what bold values in Table 3 and Table 4 represent.
* Overall, I see the benefit of the performed analysis and results showing that uncertainty measures and dice scores correlate with structure size, and there is a strong correlation between uncertainty and dice scores. However, the conclusion of the paper is not clear to me. Do authors recommend using ASSD instead of Dice (and partial correlation instead of correlation) to measure the segmentation performance when developing uncertainty measures in the future?
* The authors may want to write the conclusion after each experimental setting with a bit more clarity and motivate the necessity of the next set of experiments better. For example, at the end of Section 4.1, it is not clear what the conclusion is and how it leads to experiments in Section 4.2.
* The necessity of different (filled and empty) arrows in Table 4 is not clear. Why the analysis of how ranks differ between Table 4 and Table 5 is necessary? What do different ranks in these two tables signify?
*  Also, it would be beneficial to provide graphs (per structure for each of the correlations reported in the paper) in the appendix, as it would be easy to visualize this relationship and see the per structure effect of them.


[1] Mehta, R., Filos, A., Baid, U., Sako, C., McKinley, R., Rebsamen, M., Dätwyler, K., Meier, R., Radojewski, P., Murugesan, G.K. and Nalawade, S., 2022. QU-BraTS: MICCAI BraTS 2020 challenge on quantifying uncertainty in brain tumor segmentation-analysis of ranking scores and benchmarking results. The journal of machine learning for biomedical imaging, 2022.

[2] Camarasa, R., Bos, D., Hendrikse, J., Nederkoorn, P., Kooi, M.E., van der Lugt, A. and de Bruijne, M., 2021. A quantitative comparison of epistemic uncertainty maps applied to multi-class segmentation. arXiv preprint arXiv:2109.10702.

[3] Shaw, R., Sudre, C.H., Ourselin, S., Cardoso, M.J. and Pemberton, H.G., 2021. A Heteroscedastic Uncertainty Model for Decoupling Sources of MRI Image Quality. Machine Learning for Biomedical Imaging, 1(MIDL 2020 special issue), pp.1-23.

**Detailed Comments:**

* The first paragraph of the introduction section is missing citations for VVC, MPD, MCD, TTA and Ensembles. As these are the first instances of these methods, they require citations.
* Similarly, the last paragraph of the introduction mention GIN augmentation without citations of the full form of GIN.
* The third paragraph of the related work section ends with "by mutual information (Mukhoti and Gal 2018)". It seems out of place. Authors may want to rewrite this sentence.
* It would be great if the authors could clarify what they mean by the following sentence: "The number of annotated structures per structure type ranges from 6 (right adrenal gland) to 26 (liver), with a mean of 19."
* Authors may want to consider different uncertainty methods like conformal predictions [5] and evidential deep learning [6] in future work.

[5] Angelopoulos, A.N. and Bates, S., 2021. A gentle introduction to conformal prediction and distribution-free uncertainty quantification. arXiv preprint arXiv:2107.07511.

[6] Li, H., Nan, Y., Del Ser, J. and Yang, G., 2023. Region-based evidential deep learning to quantify uncertainty and improve robustness of brain tumor segmentation. Neural Computing and Applications, 35(30), pp.22071-22085.

**Justification Of Final Rating:**

The paper tackles an important research problem. During the rebuttal phase, the authors adequately addressed many of the concerns raised, which helped in improving the quality of the paper. Overall, I am happy to accept the paper in its current form.

**Justification Of The Preliminary Rating:**

Overall, I like the experimental study. However, the conclusion and the recommendation of the study are not clear to me. If the authors can clarify this concern in the rebuttal phase then I would be happy to accept the paper.

**Questions To Address In The Rebuttal:**

Please address concerns raised in the weakness and detailed comment section.

---

> ### Author Response · Authors · 2024-03-17
> **We thank you for your valuable feedback and adressed it in our draft as follows:**
>
> > Missing citations in the relevant work section [1] [2] [3]
> >
> > It took a while to understand that the MR data was only utilised for testing purposes. Maybe the authors want to clarify this further.
> >
> > It is not clear what bold values in Table 3 and Table 4 represent.
> >
> > The authors may want to write the conclusion after each experimental setting with a bit more clarity and motivate the necessity of the next set of experiments better. For example, at the end of Section 4.1, it is not clear what the conclusion is and how it leads to experiments in Section 4.2.
>
> We incorporated the missing literature into our paper and clarified the parts and sections you pointed out as hard to understand or suboptimally phrased, cf. the description of which data is used for training and evaluation and the conclusion of section 4.1 and motivation for section 4.2.
>
> ---
>
> > The necessity of different (filled and empty) arrows in Table 4 is not clear. Why the analysis of how ranks differ between Table 4 and Table 5 is necessary? What do different ranks in these two tables signify?
>
> Regarding the analysis of rankings in Table 4 and 5: Oftentimes, an assessment like the correlation of uncertainty with segmentation quality is used to decide which uncertainty method works best. If we switched from the "normal" correlation to partial correlation and these rankings stayed the same, then using the partial correlation would not influence the conclusions drawn from assessment based on regular correlation measures. However, we observed that these rankings do change, which shows that the conclusions drawn from partial correlations can be indeed different ones (and we argue that they are also better ones, as they remove the influence of the counfounding variable).
>
> ---
>
> > It would be great if the authors could clarify what they mean by the following sentence: "The number of annotated structures per structure type ranges from 6 (right adrenal gland) to 26 (liver), with a mean of 19."
>
> This sentence was referring the fact how many annotation masks were available for the different structure types, as the annotation process is progressing and the number of masks differs for all the structures. We now removed this sentence and included a table for the number of masks in the appendix.

---

> > ### Comment · Reviewer_hMjP · 2024-03-19
> > **Initial response**
> >
> > Could you please provide a PDF where all the changes are highlighted in a different colour? In the current state it is really difficult to see where the changes are made without access to the previous version of the paper.

---

> > > ### Author Response · Authors · 2024-03-21
> > >
> > > Dear Reviewer, please find the changes between the initial submission and our current draft here: https://owncloud.fraunhofer.de/index.php/s/selaS0qOZz4Xi5G

---

> > > > ### Comment · Reviewer_hMjP · 2024-03-21
> > > > **Reponse to Rebuttal**
> > > >
> > > > Overall, I am really happy with the rebuttal provided by the authors. It greatly improved the paper and made the reading experience much better.
> > > >
> > > > As per my understanding, the conclusion made by the authors is that Dice is greatly affected by the structure size, and as such, the correlation between dice and uncertainty is also affected by it. In this case, using the Partial correlation coefficient would somewhat circumvent the issue. Distance-based ASSD metric is less affected by the size of the structure, and as such, the correlation between ASSD and uncertainty is almost similar irrespective of the chosen correlation calculation method.
> > > >
> > > > Considering all this, I would be happy to provide acceptance of the paper.

---

### Official Review · Reviewer_EALr · 2024-03-04

**Confidence:** 4
**Preliminary Rating:** 3
**Recommendation:** Poster
**Final Rating:** 3.5

**Summary:**

The authors notice that the dice score is not independent of the size of the structure to be segmented, this is, that a model could achiveve lower dice scores just becasue the foreground is small and not because it is bad at segmenting. Then, since we might agree that uncertainty measures should in some way be inversely correlated to segmentation accuracy, it might make sense if structure size is also a confounder for uncertainty quality, in the same way as it is for segmentation.

**Strengths:**

- I don't think the relation between structure size and uncertainty quality has been studied yet, most likely because I don't think we have still arrived to a common idea of what a good uncertainty map should look like, with lots of entanglement of aleatoric and epistemic concepts.
- The analysis is done in 3d instead of in simplified 2d scenarios like the usual susptects like skin or retinal images, which is commendable.
- Good statistical analysis (using Bonferroni correction and all). In general, it is noticeable that the authors have good statistical command.

**Weaknesses:**

- Dice score measures the overlap of a prediction and the manual ground-truth. On the other hand the authors compute voxel-wise uncertainty maps, and then average over voxels predicted as the structure they are analyzing. I wonder if we are not neglecting the "False Negatives"? I mean, what happens with pixels where the model is very uncertain but predicts background and not structure? We don't care about those?
- Figure 1 does a very poor job in motivating the paper. The main idea is nice, but then this figure has lots of white space, some graphs with a very small fontsize and a caption that does not inform much about what we are seeing here. I find this to be a missed opportunity to engage the reader, by showing an example of a structure that is large or small, and the impact of that into uncertainties being judged as bad when they are actually good.
- I find the use of the partial spearman correlation coefficient a bit esoteric (and it uses R, which is a bit out of our world in medical computer vision). May the authors clarify why the SRCC does not work for them? Did they try and they did not see meaningful results?
- "we propose to use the partial correlation coefficient that controls for the structure volume when correlation segmentation quality and model uncertainty." I don't understand this sentence.

**Detailed Comments:**

What I wrote above were my detailed comments.

**Justification Of Final Rating:**

As I said, the paper is interesting and has been somehow improved during rebuttal, although the problem of ignoring false positives is still there and the lack of public python implementation hinders the adoption of the metric that is robust to confounding. Borderline acceptance sounds like a reasonable final rating, I would not be bothered if the paper is accepted but I am not going to fight for it.

**Justification Of The Preliminary Rating:**

I feel the paper is interesting and the question it poses is relevant: the way we are measuring success in building uncertainty maps might depend on structure size. On the other hand, I think that some extra explanations on what is the resulting conclusion and what should we do once we know there is a correlation between uncertainty quality and size?

**Questions To Address In The Rebuttal:**

- Even if the research question is interesting and meaningful, I end the paper not having very clear what I should do. Like, use Hausdorff distance instead of Dice? What is the recommendation that the reader should take home? I feel this should be emphasized more in the text, and also in the abstract.
- Please please please improve Fig. 1

**Special Issue:**

No

---

> ### Author Response · Authors · 2024-03-17
> **We thank you for your valuable feedback and adressed it in our draft as follows:**
>
> > Dice score measures the overlap of a prediction and the manual ground-truth. On the other hand the authors compute voxel-wise uncertainty maps, and then average over voxels predicted as the structure they are analyzing. I wonder if we are not neglecting the "False Negatives"? I mean, what happens with pixels where the model is very uncertain but predicts background and not structure? We don't care about those?
>
> Regarding the averaging of voxels over the structure mask for structure-wise epistemic uncertainty: We are aware and agree that this approach has the weakness you stated. We tried dilated masks to catch more potential false negatives, but this lead to similar results and introduces a hyperparameter of the dilation kernel size, which is why we decided against it for the sake of simplicity. As our image volumes are very large (whole body examinations) and each structure is only a very small part of it, we did not find or develop a better strategy to cope with it yet. Nevertheless this problem is to some degree adressed by the MPD and VVC as they act directly on the sets of predicted masks.
>
> ---
>
> > Figure 1 does a very poor job in motivating the paper. The main idea is nice, but then this figure has lots of white space, some graphs with a very small fontsize and a caption that does not inform much about what we are seeing here. I find this to be a missed opportunity to engage the reader, by showing an example of a structure that is large or small, and the impact of that into uncertainties being judged as bad when they are actually good.
>
> We improved figure 1 (and revamped its caption), which was rightfully and thankfully pointed out by multiple reviewers not to have conveyed our main contribution before.
>
> ---
>
> > I find the use of the partial spearman correlation coefficient a bit esoteric (and it uses R, which is a bit out of our world in medical computer vision). May the authors clarify why the SRCC does not work for them? Did they try and they did not see meaningful results?
>
> Why Spearman's rank correlation coefficient (SRCC) is not sufficient is meant to be the main message of our paper: As the segmentation quality and structure-wise uncertainty both heavily correlate with the structure size, the correlation between segmentation quality and structure-wise uncertainty gets overestimated and captures strongly the relationship of both variables with the structure size and not with each other. Therefore we propose to use partial correlation coefficients, which reduce the influence of the confounding variable on the correlation estimate between the two variables of interest. We are using the R implementation as it was the only implementation of the partial Spearman correlation introduced by Liu et al. (2018), but as our research motivates its use in this context it could make sense to re-implement it in other frameworks of course. Using the official implementation for now prevented us from making mistakes when implementing it ourselves.
>
> ---
>
> > Even if the research question is interesting and meaningful, I end the paper not having very clear what I should do. Like, use Hausdorff distance instead of Dice? What is the recommendation that the reader should take home? I feel this should be emphasized more in the text, and also in the abstract.
>
> Based on your feedback, we concluded that our contributions did not become clear enough before, and we improved the draft in many places to better reflect our main message and recommendations to the reader.

---

> > ### Comment · Reviewer_EALr · 2024-03-21
> >
> > Ok, I had a look at the revision you made available below and it seems the paper improved noticeably. Sorry for asking a silly question, and thanks for clarifying.
> >
> > I still think that the impact of your work would be greater if you were able to release a python version of the partial SRCC, and I find the answer about how false negatives are disregarded a bit like "we do not have a solution for that problem, sorry", which is not super-satisfying. It seems to me that the fair thing to do here is to upgrade my rating to borderline-accept. Good luck!

---

### Meta-Review · Area_Chair_hjiF · 2024-04-03

**Recommendation:** Accept (Poster)
**Confidence:** 4

**Metareview:**

The paper proposes to investigate structure size as a confounder variable in utilizing uncertainty estimates for segmentation quality prediction. It tackles an important research problem.
During the rebuttal phase, the authors adequately addressed many of the concerns raised, which helped in improving the quality of the paper.

---

### Decision · Program_Chairs · 2024-04-05

Accept (Poster)